# Short hydrogen-bond network confined on COF surfaces enables ultrahigh proton conductivity

Benbing Shi[1,7], Xiao Pang[1,7], Shunning Li [2,7], Hong Wu [1,3], Jianliang Shen [1], Xiaoyao Wang [1], Chunyang Fan[1], Li Cao[1], Tianhao Zhu[1], Ming Qiu[1], Zhuoyu Yin[1], Yan Kong[1], Yiqin Liu[1], Mingzheng Zhang[2], Yawei Liu[4], Feng Pan [2] ✉ & Zhongyi Jiang [1,3,5,6] ✉

The idea of spatial confinement has gained widespread interest in myriad applications. Especially, the confined short hydrogen-bond (SHB) network could afford an attractive opportunity to enable proton transfer in a nearly barrierless manner, but its practical implementation has been challenging. Herein, we report a SHB network confined on the surface of ionic covalent organic framework (COF) membranes decorated by densely and uniformly distributed hydrophilic ligands. Combined experimental and theoretical evidences have pointed to the confinement of water molecules allocated to each ligand, achieving the local enrichment of hydronium ions and the concomitant formation of SHBs in water-hydronium domains. These overlapped water-hydronium domains create an interconnected SHB network, which yields an unprecedented ultrahigh proton conductivity of 1389 mS cm$^{-1}$ at 90 °C, 100% relative humidity.

Proton transfer is a ubiquitous process in green energy storage and conversion devices, such as fuel cells and flow batteries[1–3]. Creating artificial ion channels that confer high proton conductivity has long been an aspirational endeavor for improving their power output[4–6]. As one of the promising strategies to promote proton conductivity, spatial confinement of water molecules in a laminated/porous architecture has been recently explored, which can yield a proton-rich local environment, leading to higher concentration of charge carriers and more rapid water-mediated proton transfer[7–14]. It is well recognized that the proton transfer in aqueous systems is governed by the Grotthuss mechanism, which takes place on a water-hydronium network and entails a collective proton motion akin to the Newton's cradle[15–17]. Of paramount importance in understanding

the Grotthus-type proton conduction is that its efficiency mainly relies on the state of the proton bridges interconnected by hydrogen bonds (H-bonds). Commonly, the intensity of the H-bonds is associated with the distance between the oxygen atoms of the hydrogen-bonded water molecules[18], and stronger H-bonds would lead to a more delocalized nature of the protons between donor and acceptor molecules. In particular, short H-bonds (SHBs) with a donor-acceptor distance of below 2.5 Å could generate a superharmonic behavior of proton motion, triggering proton transfer in a nearly barrierless manner[19]. The confinement of water in either bulk or interfaces of materials can serve as a strategy to favor the formation of a SHB network[20], since the excess protons are accommodated in a more constrained space than normal. However, its implementation is still

[1]Key Laboratory for Green Chemical Technology of Ministry of Education, School of Chemical Engineering and Technology, Tianjin University, 300072 Tianjin, China. [2]School of Advanced Materials, Peking University Shenzhen Graduate School, 518055 Shenzhen, Guangdong, China. [3]Haihe Laboratory of Sustainable Chemical Transformations, 300192 Tianjin, China. [4]Beijing Key Laboratory of Ionic Liquids Clean Process, CAS Key Laboratory of Green Process and Engineering, State Key Laboratory of Multiphase Complex Systems, Institute of Process Engineering, Chinese Academy of Sciences, 100190 Beijing, China. [5]Joint School of National University of Singapore and Tianjin University, International Campus of Tianjin University, Binhai New City, 350207 Fuzhou, China. [6]Zhejiang Institute of Tianjin University, 315201 Ningbo, Zhejiang, China. [7]These authors contributed equally: Benbing Shi, Xiao Pang, Shunning Li. ✉e-mail: panfeng@pkusz.edu.cn; zhyjiang@tju.edu.cn

challenging due to the difficulty in satisfying the connectivity of the network when the SHBs are dispersedly distributed.

One prerequisite for water confinement at material surface resides in the appropriate hydrophilic ligands, among which the sulfonic acid group (-SO$_3$H) was recognized as a superior proton donor and an enabler for high proton conductivity[21–24]. We may conjecture that the concentrated hydronium ions in confined water around -SO$_3^-$ could induce the rearrangement of water molecules with a high propensity of SHB formation in water-hydronium domains (WHD, H$_2$O·H$^+$·OH$_2$) (Fig. 1a). In comparison to the facile proton transfer inside a confined water domain and between over-lapped domains to share WHD, the migration of protons between adjacent descrete domains is expected to be analogous to that with normal H-bonds, which involves a relatively higher barrier (Fig. 1b). To mitigate the proton transfer outside the confined water domains, an essential key lies in the establishment of a highly interconnected architecture of these water domains. For proof of concept, we capitalize on the recent development of reticular chemistry[25–27] and focus on a set of ionic covalent organic framework membranes (iCOFMs) decorated with -SO$_3$H groups. The robust, regular crys-talline skeleton of iCOFMs[28–31] affords an ideal platform to uncover the proton transfer behavior featured by SHBs.

## Results and discussion

In this work, the iCOFMs with controllable -SO$_3$H group distance were de novo designed and synthesized based on modular engineering and vacuum-assisted self-assembly method. The -SO$_3$H groups display an ordered distribution, building up a precisely regulated WHD with concentrated hydronium ions and a fully interconnected H-bond net-work via the overlap to share WHD between each pair of neighboring domains (Fig. 1c). The perfectly uniform alignment of these domains can circumvent the occasional disruption of H-bond network as observed in other proton exchange membranes. This disruption is also observed in loosely packed iCOFMs, and their inferior performance to the tightly packed counterparts confirms the essential role of an integral SHB network (Fig. 1d). We justify that the short H-bonds emerge in the water domains around -SO$_3^-$, and find that the limited number of water molecules not only imparts a high concentration of hydronium ions but also defines a critical region for proton transfer. The intersection of these regions as imposed by the small distance between neighboring -SO$_3$H groups could endow the membrane with an ultrahigh proton conductivity of 1389 mS cm$^{-1}$ at 90 °C, 100% RH. We demonstrate that a high proton conductivity is maintained even at 40% RH, which could be ascribed to the self-adaptation of the surface-confined H-bond network. We believe that these mechanistic insights could be extended to other materials or systems, and leveraged for the rational design of advanced proton exchange membranes.

The topological structures of the iCOFs are shown in Fig. 2a. iCOFs unit with six -SO$_3$H groups represents the larger group distance ~2.0 nm and iCOFs unit with twelve -SO$_3$H groups represents the shorter group distance ~0.9 nm. The -SO$_3$H groups are fixed on the iCOF skeleton and the group distance can be regulated from 2.5 to 0.8 nm based on the crystal parameter data (Fig. 2b, Supplementary Figs. 1 and 2, and Tables 1–6). Single-phase and two-phase synthesis methods were exploited to control the monomer diffusion and che-mical assembly processes to fabricate crystalline iCOF nanosheets (Supplementary Fig. 3). Different synthesis conditions were adopted based on the reactivity and solubility of amine monomer and aldehyde monomer (Supplementary Table 7 and Figs. 4 and 5). The transparent iCOF nanosheets suspensions were obtained after reacting for three to twenty days and exhibited obvious Tyndall phenomenon (inset of Fig. 2c and Supplementary Fig. 6). As shown in Supplementary Fig. 7, the zeta potential value of iCOF nanosheets was positively correlated with the IEC value. Due to the strong electrostatic repulsion between iCOF nanosheets, the suspensions kept stable for two years and no flocculation or precipitation was observed. The iCOF nanosheets bore flake-like morphology and size of several micrometer with thickness of 1.6 nm (Supplementary Fig. 8). High-resolution TEM characterization of iCOF nanosheet reveals that the long-range ordered lattice fringe spacing is about 0.34 nm, which is in agreement with the (001) facet (Supplementary Fig. 9)[32,33]. The functional groups and chemical structure of the iCOF nanosheets were characterized by FTIR, XPS, and NMR (Supplementary Figs. 10–12). The iCOF nanosheets exhibit high-thermostability up to 300 °C, well satisfying the practical requirement of electrochemical energy storage and conversion devices (Supple-mentary Fig. 13).

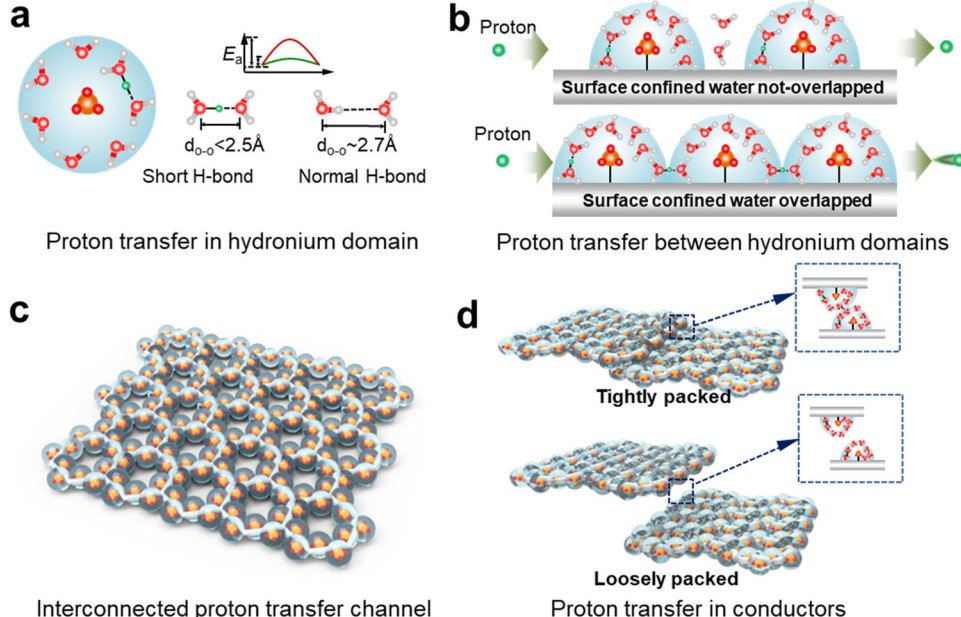

**a** Proton transfer in hydronium domain

**b** Proton transfer between hydronium domains

**c** Interconnected proton transfer channel

**d** Proton transfer in conductors

**Fig. 1 | Schemes of proton transfer in confined water. a** Proton transfer via SHB and normal H-bond in a confined water domain. **b** Proton transfer between two separated water domains and between two overlapped water domains shared with SHB, respectively. **c** Establishment of interconnected SHB network for proton transfer, achieved by ordered distribution of surface -SO$_3$H groups. **d** Scenario of descrete water domains with inferior proton conductivity.

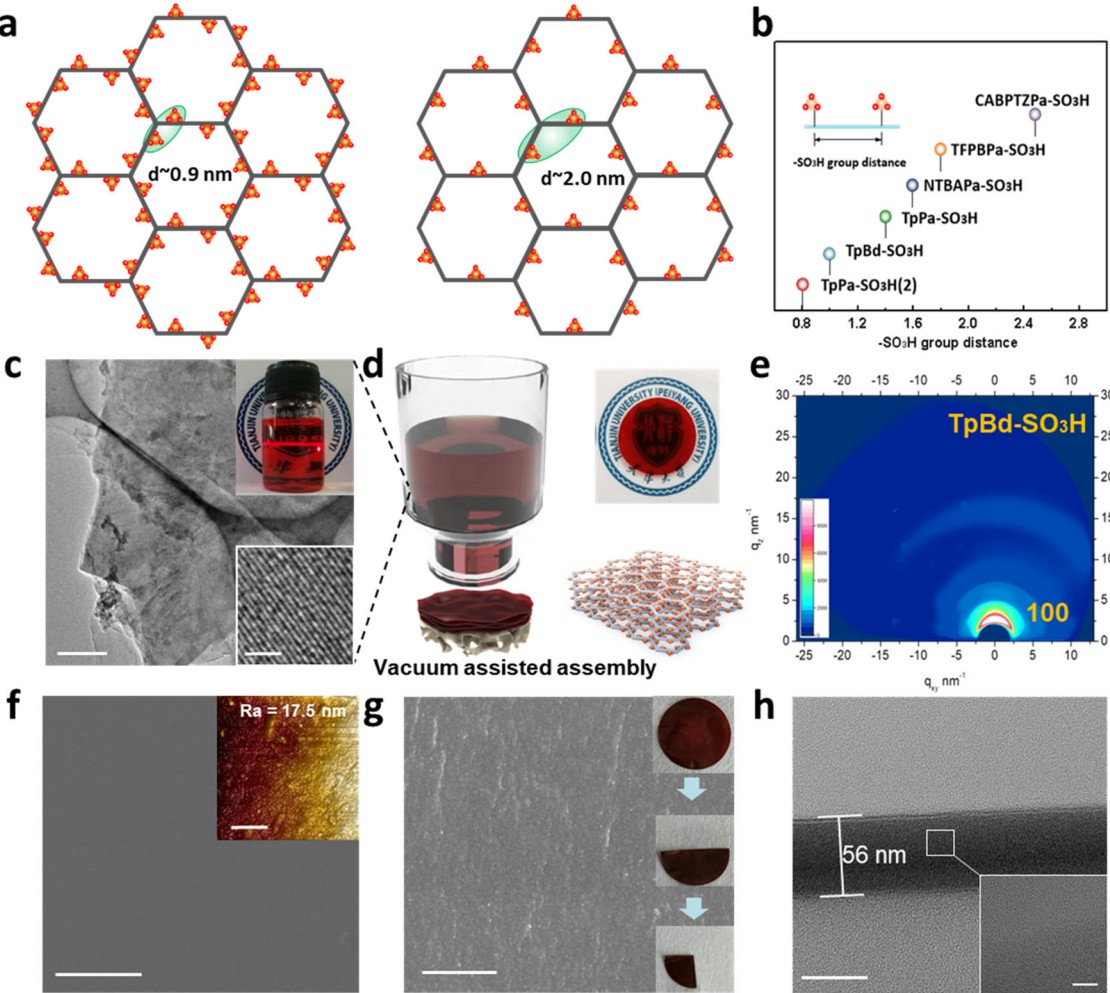

**Fig. 2 | Preparation and characterizations of iCOF nanosheets and iCOFMs.**
**a** Topological structures of the iCOFs with twelve and six -SO₃H groups on each hexagonal unit, respectively(the green ellipse represents the -SO₃H groups distance). **b** Scheme of the -SO₃H group distance in iCOFs skeleton. **c** TEM image of the iCOF nanosheets (scale bar, 500 nm, 2 nm). The upper and lower insets display the photograph of the iCOFs nanosheet solution and the HRTEM image of iCOFs nanosheets, respectively. **d** Scheme of the preparation of iCOFM. The upper and lower inserts display the photograph and laminated structure of iCOFM, respectively. **e** The GIWAXs data of TpBd-SO₃H. **f** SEM image of iCOFM (scale bar, 10 μm, 1 μm). The insert shows the AFM image of iCOFM. **g** SEM image of cross-section of the iCOFM (scale bar, 1 μm). The inserts show the photographs of folded iCOFM. **h** TEM image of the cross-section of the iCOFM (scale bar, 50 nm). The inset shows the magnified image (scale bar, 2 nm). Source data are provided as a Source Data file.

The iCOFMs were prepared by vacuum-assisted self-assembly method, as shown in Fig. 2d, Supplementary Figs. 14 and 15. The iCOFMs are transparent with high crystallinity, mechanical robustness, and remain stable in water, DMF, and 1 M H₂SO₄ (60 °C) solution for more than two years, which can be attributed to the strong covalent and π-π interactions between high crystalline nanosheets (Supplementary Figs. 16–18)[34,35]. Moreover, the iCOFMs display arcs of diffracted intensity respect to (100) facet in the 2D GIWAXs spectrum, manifesting that the iCOF nanosheets are almost parallely aligned (Fig. 2e and Supplementary Fig. 19)[36,37]. The surface of iCOFMs is smooth, as observed from the surface SEM image (Fig. 2f). Besides, the strong π-π interaction endows the membrane with compact structure and the membrane keeps intact after being folded for several times (Fig. 2g). No interlayer defects are observed even when the thickness of the membrane decreases to 56 nm (Fig. 2h). Meanwhile, the membrane exhibits in-plane swelling ratio <5% in water with the increase of temperature due to the synergy of covalent, π-π, and electrostatic interactions (Supplementary Fig. 20). The ion exchange content (IEC) values of the iCOFMs were determined by acid-base titration method, as shown in Supplementary Fig. 21. The IEC value significantly increase with the decrease of the -SO₃H group distance and a value of

5.4 mmol g⁻¹ was achieved, which is the highest IEC values ever reported.

The in-plane proton conductivity of the iCOFMs was measured by two electrode method and calculated according to the resistance (Supplementary Fig. 22), and the proton transfer pathway was illustrated in Fig. 3a. In-plane proton transfer is along the surface groups on the iCOF skeleton. As the -SO₃H group distance decreases from 1.4 (TpPa-SO₃H) to 1.0 nm (TpBd-SO₃H), the proton conductivity increases six folds. The highest proton conductivity reaches 1389 mS cm⁻¹ at 90 °C under 100% RH when the group distance decreases to 0.8 nm (TpPa-SO₃H(2), Supplementary Fig. 23), which is the highest proton conductivity ever reported. In sharp contrast, when the group distance is >1.4 nm, the iCOFM displays much lower proton conductivity of <0.2 S cm⁻¹ in spite of the high IEC value (3.42 mmol g⁻¹). Moreover, the proton conductivity remains almost unchanged when further increasing or decreasing the -SO₃H group distance. Note that the proton conductivity significantly decreased as the iCOF nanosheets (d–1.0 nm) are loosely packed as characterized by SEM (Fig. 1d, Supplementary Figs. 24 and 25). The water vapor adsorption results indicate that both membranes display similar hydration number (λ) of 6.5 and 6.8 at 30 °C under 100% RH (Fig. 3b), corresponding to the average

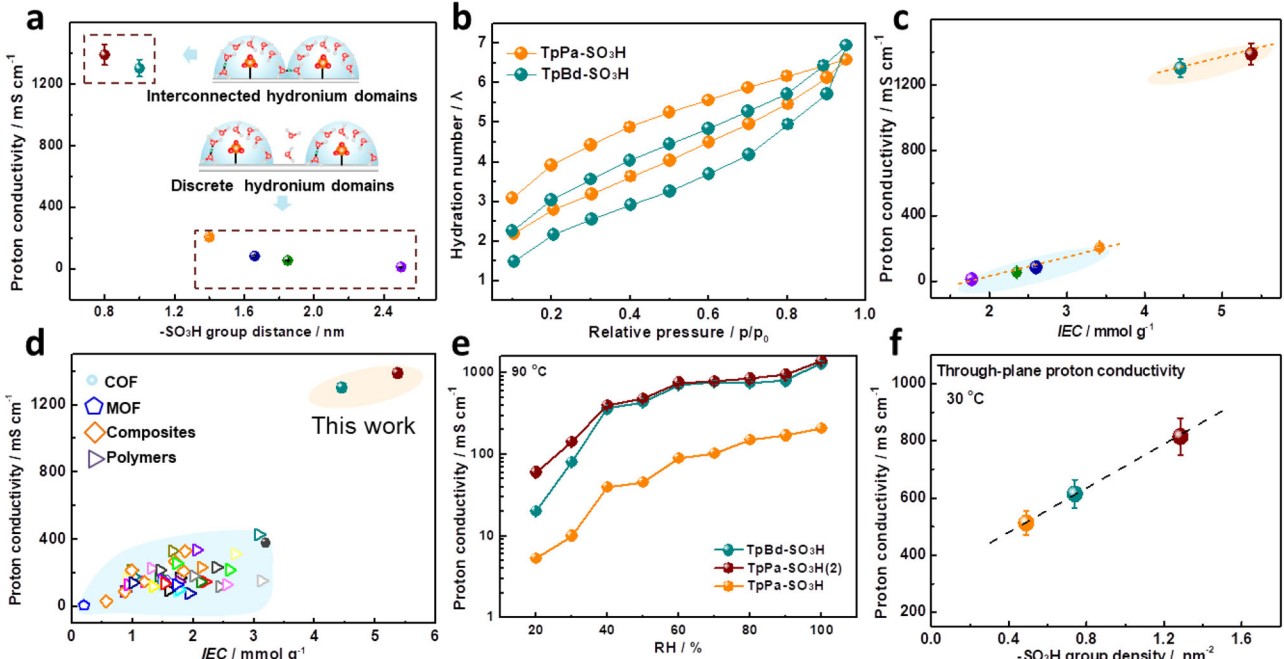

**Fig. 3 | In-plane and through-proton conductivity of the iCOFMs. a** -SO₃H group distance dependent proton conductivity of the iCOFMs at 90 °C under 100% RH (inset is the scheme of the in-plane proton transfer on iCOFMs). **b** Relative humidity-dependent hydration number of the TpPa-SO₃H and TpBd-SO₃H membranes at 30 °C. **c** IEC value dependent proton conductivity of the iCOFMs at 90 °C under 100% RH. **d** The comparison of the IEC value and proton conductivity between the iCOFMs and the membranes reported in literatures. **e** Relative humidity-dependent in-plane proton conductivities of the iCOFMs at 30 °C. **f** Through-plane proton conductivities of the iCOFMs at 30 °C in 1 M HCl solution. All the error bars in this figure represent the standard deviation of the experiments. Source data are provided as a Source Data file.

number of confined water molecules per -SO₃H for TpPa-SO₃H and TpBd-SO₃H, respectively. Accordingly, the distribution of surface water/hydronium will be constrained in a limited region, outside which the donor-acceptor water bridge is no longer valid[38]. The short group distance in TpBd-SO₃H gives rise to the overlap between these regions and therefore an interconnected H-bond network shared with WHD, while the group distance in TpPa-SO₃H is large enough to allow the isolation of each water domain, leading to a discrete H-bond network.

According to the Nernst-Einstein equation ($\sigma = DCF^2/RT$), the proton diffusion coefficient (D) in TpBd-SO₃H is 4.48 folds higher than that of TpPa-SO₃H. Moreover, the proton conductivity was normalized by IEC value to exclude the influence of the amount of -SO₃H group on proton transfer (Supplementary Fig. 26). It is found that the -SO₃H group distance plays vital role in proton transfer, especially when the -SO₃H group distance decreases from 1.4 nm to 1.0 nm (Fig. 3c). For -SO₃H group with specific position, localized WHD forms in water domain around -SO₃H group, in which proton shuttles with a nearly barrierless manner. As the -SO₃H groups distance decrease to 1.0 nm, the water domains are overlapped and the WHDs can be shared among -SO₃H groups on iCOF nanosheet surface. When the iCOF nanosheets are tightly packed, WHDs can be shared between iCOF nanosheets, which enables ultrafast proton transfer in iCOFM. Furthermore, benefiting from the covalent, π-π, electrostatic interactions, as well as high IEC values, our iCOFMs acquired significant superiorities in overcoming the trade-off between proton conductivity and mechanical stability, as compared with the membranes reported in literatures (Fig. 3d, Supplementary Table 8, and Fig. 27).

The relative humidity (RH) dependent in-plane proton conductivities of the iCOFMs at 30 °C and 90 °C were shown in Fig. 3e and Supplementary Fig. 28. The TpPa-SO₃H(2), TpBd-SO₃H, and TpPa-SO₃H exhibit lower proton conductivity with the decrease of RH at both 30 °C and 90 °C due to the reduction of the hydration number. Notably, the proton conductivities of TpPa-SO₃H(2), TpBd-SO₃H maintain at ~10⁻¹ S cm⁻¹ even when the RH decreases to 40%, while

TpPa-SO₃H displays inferior proton conductivity of about 10⁻³ S cm⁻¹, indicating that the surface confinement endows the H-bond network with high flexibility for self-adaption rearrangement, thus facilitating the preservation of high proton conductivity at low RH. This finding can help accelerate the exploitation of weakly humidity-dependent proton exchange membranes.

The through-plane proton conductivities of the iCOFMs were measured by sandwiching the iCOFMs between two 1 M HCl solutions. The proton conductivity of iCOFMs can be derived based on the I-V curves, the thickness and effective area of the membranes (Supplementary Fig. 29). In the through-plane direction of iCOFMs, the short -SO₃H group distance (0.68 nm) and straight proton transfer pathway endow the iCOFMs with proton conductivity of hundreds of mS cm⁻¹ at 30 °C, which is several times higher than that of Nafion 117 (Supplementary Fig. 30). Moreover, the through-plane proton conductivity of iCOFMs is positively correlated with -SO₃H group density (Fig. 3f). Finally, the TpPa-SO₃H and TpBd-SO₃H nanosheets were packed into membrane electrode assemblies (MEAs) to evaluate the potential applications in fuel cells (Supplementary Fig. 31). Both of the MEA exhibit open circuit voltage (OCV) of 0.95 V (theoretical value, 1.22 V) ascribed to higher gas permeability. However, the maximum power density and current density of TpBd-SO₃H reach about 423.4 mW cm⁻² and 1280 mA cm⁻², which can be attributed to the higher through-plane proton conductivity. Despite the high gas permeability, the iCOFMs exhibit comparable fuel cell performance than that of Nafion 117 under the same condition. Further optimizing the topological structure of iCOFMs based on the surface-confined concentrated hydronium ions would be an efficient way to simultaneously facilitate the gas barrier property, proton conductivity, and fuel cell performance.

The contribution of channel surface groups to proton transfer was further explored by experiment and molecular simulation. The absorbed water in iCOF nanochannel includes bound water and free water according to DSC and molecular simulation results

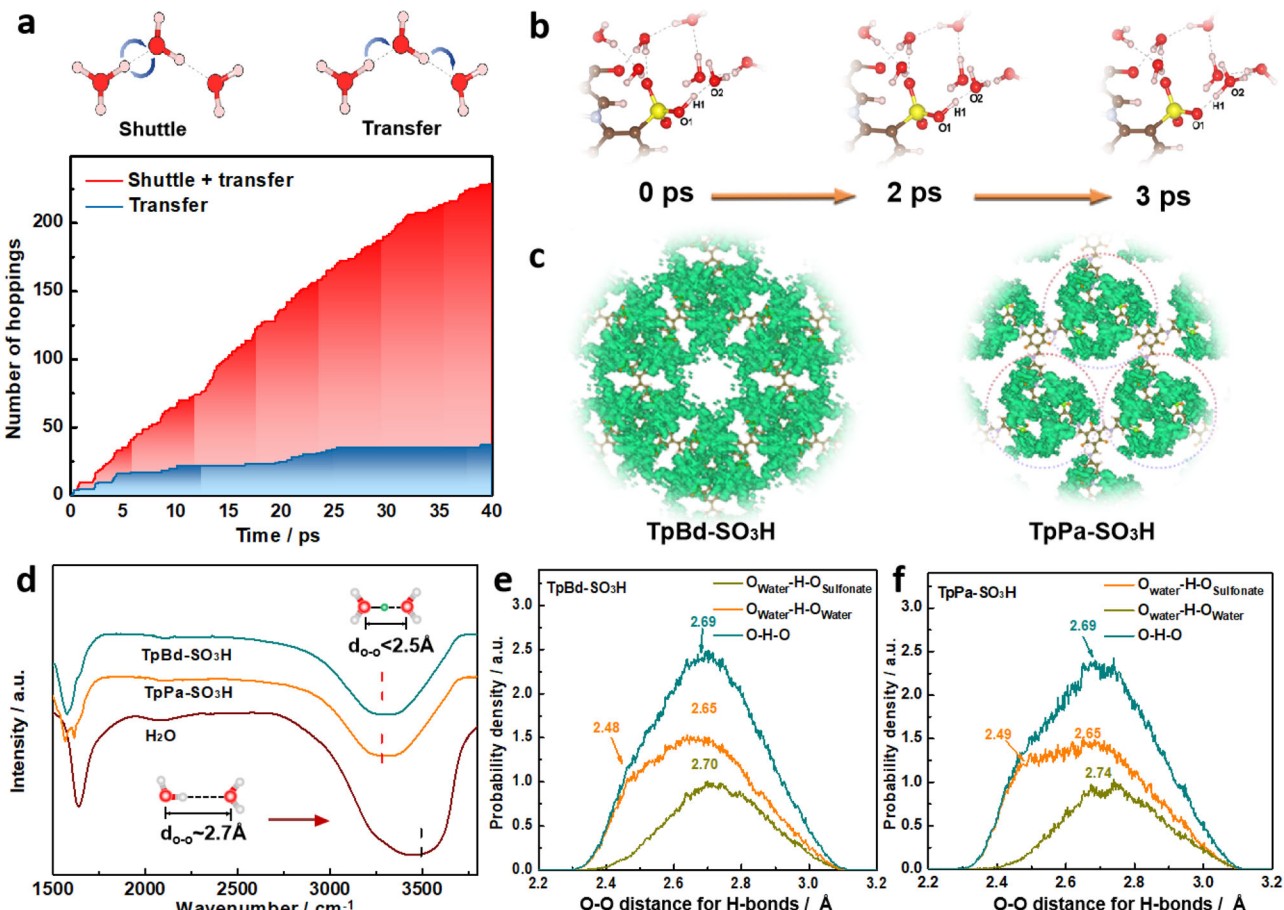

**Fig. 4 | Proton transfer mechanism in iCOFMs. a** Schematics of a shuttle event and a transfer event, and the number of proton hopping as a function of the simulation time. **b** Facile proton dissociation behavior. **c** The probability density distribution of $H_3O^+$ in TpBd-SO$_3$H and TpPa-SO$_3$H. **d** FTIR spectrum of water and wet TpBd-SO$_3$H. **e**, **f** O-O distance in water and wet TpBd-SO$_3$H and TpPa-SO$_3$H from AIMD simulations. Source data are provided as a Source Data file.

(Supplementary Figs. 32 and 33). Due to the strong molecular affinity between $H_2O$ and -SO$_3$H group, water molecule forms a denser adsorption layer (i.e. hydration shell) near the -SO$_3$H group, whose outer edge is away from the central S atom by 0.5 nm (Supplementary Fig. 34). When the -SO$_3$H group distance is set as 2.0 nm, the adsorption layers are almost isolated. However, as the -SO$_3$H group distance decrease to 1.0 nm, the hydration shells are overlapped. The water molecule distribution in iCOFs nanochannel was obtained by molecular simulation, as shown in Supplementary Figs. 35 and 36. The results demonstrate that the water molecule density on channel surface is much higher than that in bulk water, which confers high hydrogen bond density on channel surface. Moreover, the water molecules along channel surface keep frequent exchanges with bulk water, suggesting the flexible rearrangement of the water/hydronium network (Supplementary Fig. 37)[39].

Ab initio molecular dynamics (AIMD) simulation was performed to elucidate the microscopic picture of proton transfer in surface-confined water. We first evaluate the hopping frequency of protons in the confined region of seven water molecules surrounding each -SO$_3$H. Here, the single-layer TpPa-SO$_3$H was taken as a model system. Two kinds of hopping events are discriminated: shuttle events which are associated with the consecutive to-and-from motions of a proton between two neighboring water molecules, and transfer events where the proton keeps forward motion (Fig. 4a)[40–42]. The simulation results reveal that the former event dominates in the confined region. This can be attributed to the attraction of the negatively charged -SO$_3^-$ species to protons, which prevails after the prompt dissociation of proton from the -SO$_3$H group, as verified by the short period (<3 ps) for proton

to reach an adjacent water molecule and form a hydronium ion (Fig. 4b and Supplementary Fig. 38). We note that this proton would no longer recombine to the -SO$_3^-$ species after its detachment. Given the low kinetic barrier for Grotthuss-type proton transport in water-hydronium network (Supplementary Fig. 39), the electrostatic interaction between protons and -SO$_3^-$ species will be major contributing factor for the spatial confinement of the hydronium ions[43]. Similar situation is found in $H_2SO_4$ aqueous solution (Supplementary Figs. 40 and 41). This interaction will pose hindrance to the transport of hydronium ions between two adjacent separate water domains.

In comparison to the pattern of isolated water domains on TpPa-SO$_3$H, the confined regions on TpBd-SO$_3$H is interconnected with each other, as revealed by the probability density distribution of hydronium ions displayed in Fig. 4c. The establishment of the interconnected water-hydronium network permits facile Grotthuss-type proton transport between the confined regions, which is reminiscent to the π-electron delocalization in conjugated systems[44–46]. The overlap of the confined regions also leads to a more vague boundary for each of these regions, as illustrated by the more prevalent smearing of peaks in the radial distribution functions between O in hydronium ions and S in the -SO$_3$H species (Supplementary Fig. 42).

Owing to the electrostatic interactions, protons quickly shuttle between the surface-confined water molecules rather than transport to bulk water, thus affording a high concentration of hydronium ions in the H-bond network. This feature may benefit the emergence of SHBs as each hydronium ion tends to engage in a SHB with one of its neighboring water molecules[47]. FTIR was utilized to characterize the H-bonds in bulk water, acid solution, and surface confined water in

iCOFM (Fig. 4d). Pure water exhibits absorption bond at around 3500 cm$^{-1}$ ascribed to the stretching vibration of H-bond. By comparison, the absorption bond shifts to 3200 cm$^{-1}$ in H$_2$SO$_4$ solution and wet iCOFM attributed to emergence of SHBs (Supplementary Fig. 43). This is in good agreement with the AIMD results of the distribution of O-O distance for all the H-bonds (Fig. 4e and Supplementary Fig. 44). The SHB allows proton sharing between donor and acceptor, which renders low proton transfer resistance[48]. We note that 44% H$_2$SO$_4$ aqueous solution possesses approximately one hydrated proton for every seven water molecules and exhibits a proton conductivity on the scale of ~1600 mS cm$^{-1}$ at 90 °C (Supplementary Fig. 45). The proton conductivity of TpBd-SO$_3$H is close to this value, suggesting that the essence of facile proton transfer in surface-confined water lies in the local concentration of hydronium ions and SHBs. The overlap of water confined regions triggers the delocalization of protons such that a fast proton transfer network is built on the surface. Based on the above results, the prerequisite for this delocalization is a short distance between the hydrophilic groups (no larger than 1.0 nm). The proton transfer isotope effect of the iCOFMs was probed by measuring the proton conductivity under H$_2$O or D$_2$O vapor, respectively. The $\sigma_{H_2O}/\sigma_{D_2O}$ ratios fall into the range of 1.6–2 and the activation energy ($E_a$) of iCOFMs is less than 10 kJ mol$^{-1}$, indicating that the proton transfer in the surface confined water regions obeys the Grotthuss mechanism (Supplementary Figs. 46 and 47)[49–52].

In summary, a strategy of constructing a surface-confined SHB network via a precisely regulated alignment of hydrophilic ligands was proposed to develop high-performance proton exchange membranes. Crystalline iCOFMs with tunable group distance were de novo designed and fabricated, which exhibited a maximum proton conductivity of 1389 mS cm$^{-1}$. It is uncovered that a limited amount of water molecules are confined by each -SO$_3$H with its proton detached and binding to an adjacent water molecule. The confinement of water molecules ensures a high local concentration of hydronium ions and SHBs in each WHD, which also necessitates a short group distance to enable the establishment of an interconnected SHB network. Once established the network would yield an ultrahigh proton conductivity that can be retained even at low RH. This ordered architecture of surface-confined water domains affords new design paradigm where facile proton transfer and weak dependence of humidity are ingeniously incorporated.

## Methods
### Materials and reagents
2, 5-diaminobenzene sulfonic acid (Pa-SO$_3$H), 2,5-diaminobenzene-1,4-disulfonic acid (Pa-SO$_3$H(2)), 1,3,5-trihydroxy homobenzaldehyde (Tp), 4,4′,4″-nitrilotribenzaldehyde (NTBA), 5′-(4-formylphenyl)-[1,1′:3′,1″-terphenyl]−4,4″-dicarbaldehyde (TFPB), and 4′,4‴,4″″-(1,3,5-triazine-2,4,6-triyl)tris([1,1′-biphenyl]−4-carbaldehyde)) (CABPTZ) were purchased from Jilin Chinese Academy of Sciences-Yanshen Technology Co., Ltd. (Jilin, China). 4′-diamino-[1,1′-biphenyl]−3,3′-disulfonic acid (Bd-SO$_3$H) was obtained from Henan Alpha Technology Co. Ltd. Octanoic acid, mesitylene, N, N-dimethylformamide (DMF), o-dichlorobenzene1, and 1,4-dioxane were purchased from Aladdin Reagents Co., Ltd. All the reagents were used as obtained without further purification. Deionized water (18.2 MΩ) was used throughout the experiment.

### Synthesis of ionic covalent organic framework nanosheets
The ionic covalent organic framework (iCOF) nanosheets were prepared by single or two-phase synthesis method. Ionic amine monomer (Pa-SO$_3$H (0.15 mmol), Pa-SO$_3$H(2) (0.15 mmol), Bd-SO$_3$H (0.15 mmol)) and aldehyde monomer (Tp (0.1 mmol), NTBA (0.1 mmol), TFPB (0.1 mmol), and CABPTZ (0.1 mmol)) were dissolved into water (20 ml), n-caprylic acid (20 ml) or dimethyl sulfoxide (20 ml), respectively. Both solutions were filtrated to remove the undissolved

monomers. The reactors were placed in constant temperature chamber for several days. The temperature was set based on the reactivity of the monomers and detailed reaction conditions were listed in Table S1. The iCOFs dispersion was filtrated and purified by dialysis for three days to remove the unreacted monomers.

### Preparation of ionic covalent organic framework membranes
The iCOFMs were prepared by vacuum-assisted assembly method. First, the iCOF nanosheets were assembled on the surface of PAN driven by vacuum pressure of −0.025 MPa and −0.08 MPa. Then, the composite iCOF membranes were immersed into DMF to peel off the PAN substrate and a freestanding iCOFM was obtained. At last, the iCOFM was placed in 1 M H$_2$SO$_4$ solution at 60 °C for 3 days.

### Characterization of iCOFMs
The chemical structure and the element contents of iCOF nanosheets and iCOFMs were characterized by FTIR, NMR, and XPS. The morphology, lattice fringe, and element distribution of iCOF nanosheets and iCOFMs were characterized by TEM (Tecnai G220 S-TWIN) and SEM (Nanosem 430). The thickness of the iCOF nanosheets and the roughness of the membranes were recorded by a Bruker Dimension FastScan. The crystallinity of iCOFMs was characterized by XRD using a Bruker D8 Advance X-ray diffractometer equipped with Cu Kα radiation (λ = 1.5406 Å) at a scanning rate of 8° min$^{-1}$. The thermal stability of iCOFMs was measured by a Shimadzu TGA-50 from 40 to 800 °C with a ramp rate of 10 °C min$^{-1}$ under nitrogen atmosphere. Stress and strain of iCOFMs were tested by using a universal tensile machine (UTM, Lloyd). The water-vapor sorption at 30 °C was obtained by monitoring the mass change of iCOFMs under various humidity.

### Swelling ratio of iCOFMs
The iCOFMs were dried at 100 °C under vacuum conditions for 24 h before testing the swelling ratio. Then, the dry iCOFMs were immersed into water at the temperature ranging from 30 to 90 °C. The swelling ratio of iCOFMs was calculated by equation as follows:

$$\text{Swelling ratio (\%)} = \frac{A_{\text{wet}} - A_{\text{dry}}}{A_{\text{dry}}} \times 100\% \tag{1}$$

where, the $A_{\text{wet}}$ and $A_{\text{dry}}$ were the area of dry and wet iCOFMs, respectively.

### IEC values of iCOFMs
The IEC values of iCOFMs were obtained by a titration method. The dry membranes (0.2 g) were immersed into NaCl solution (0.01 g/mL, 20 mL) and stirred for 24 h. Then the solutions were titrated using a 0.01 M NaOH solution, and phenolphthalein was used as indicator. The IEC values (mmol g$^{-1}$) of the samples were calculated by Eq. (2)

$$\text{IEC} = \frac{C_{\text{NaOH}} \times V_{\text{NaOH}}}{W_{\text{dry}}} \tag{2}$$

where, the $C_{\text{NaOH}}$ and $V_{\text{NaOH}}$ were the concentration and consumed volume of NaOH solution, and $W_{\text{Dry}}$ (g) was the weight of the dry iCOFMs.

### Proton conductivity of iCOFMs
The in-plane proton conductivities of iCOFMs were measured using a two-electrode ac impedance spectroscopy technique. The resistance value (Ω) was tested over the frequency range of 1 M–1 Hz with an oscillating voltage of 15 mV. Proton conductivity measurements were carried out in a thermo-controlled chamber equipped with water and D$_2$O vapor. The proton conductivities ($\sigma$, mS cm$^{-1}$) of membranes were

calculated by Eq. (3):

$$\sigma = \frac{l}{R \times A} \qquad (3)$$

where $l$ (cm), $R$ ($\Omega$), and $A$ (cm$^2$) represent the membrane thickness, resistance, and contact area, respectively.

The through-plane proton conductivities of the iCOFMs were measured by sandwiching the iCOFMs between 1 M HCl solutions. The samples were prepared by assembling the iCOFMs onto polyethylene diaphragm (100 μm) with pore size of 0.5 mm × 0.5 mm. The resistance with and without iCOFMs was both measured. The resistances of iCOFMs were obtained by removing the resistance of HCl solution from the total resistance. The through-plane proton conductivities ($\sigma$, mS cm$^{-1}$) of the iCOFMs were calculated by Eq. (3).

### Single-cell performance evaluation
Membrane electrode assemblies (MEA) were fabricated by sandwiching the iCOFMs between two catalytic electrodes without hot pressing. Then, the MEAs were sandwiched between two graphite electrodes and assembled into single cells. The flow rates of hydrogen and oxygen were both controlled at 100 mL min$^{-1}$ and 400 mL min$^{-1}$ without applying back pressure. The performance evaluation of fuel cells was performed at 60 °C under 100% RH.

### H$_2$O aggregate state in the iCOF nanochannel
The isolated -SO$_3$H group is simply represented by the rigid tetrahedral SO$_4{}^{2-}$ with one O atom being replaced by C atom. The atomic charges on S atom, O atom, C atom are +2e, −1e and 0e, respectively, so that the single -SO$_3{}^-$ group has a net charge of −1e. The Lennard-Jon//es (LJ) potential parameters were used for S-S pairs and O-O pairs and the Dreiding force field was used for C-C pairs[53,54]. The water molecules were represented by the SPC/E model[2]. The atomic structure of the TpPa-SO$_3$H nanosheets with -SO$_3{}^-$ groups from Ref. 2. was used. The atomic charges of atoms were calculated by the DFT method with the PBE0 and def2tzvp basis set using Gaussian 16. Dreiding force field was used for the LJ potential parameters. In the simulations, hydroniums (H$_3$O$^+$) were added to maintain the neutralization of the system, and the model in ref. 55. was used for H$_3$O$^+$.

All simulations were carried out using the parallel MD software package LAMMPS[56]. Periodic boundary conditions were imposed in all three directions. The velocity-Verlet algorithm with a time step of 1 fs was used to integrate the equation of motion, and a Nosé−Hoover thermostat with a time constant of 100 fs was used to maintain the temperature of the fluid at T=298 K. The Lennard-Jones (LJ) potential was cut and shifted at 1.0 nm. The electrostatic interaction was also cut at 1.0 nm and the particle−particle particle−mesh (PPPM) with an accuracy of 10$^{-4}$ was employed to calculate the long-range electrostatic interactions. The arithmetic mixing rule was applied for the LJ potential parameters between different species. The SHAKE algorithm was used to maintain the rigidity of the water and methanol molecules[57].

In the simulations of -SO$_3$H group(s) in liquid water, there were 2744 H$_2$O (plus 1 or 2 H$_3$O$^+$) in a cubic simulation box with edges of ~4 × 4 × 4 nm. -SO$_3$H groups were constrained in a fixed position by applying a harmonic potential on the central S atoms. In addition, another harmonic potential was applied on the center-of-mass of all three O atoms in the same group to maintain the orientation of the -SO$_3$H group(s) during the simulation, so that the vector from S atoms to the center-of-mass of O atoms was almost along the $z$-direction. NPT simulations were performed with a Nosé−Hoover barostat with a time constant of 1 ps applied to maintain the liquid pressure at 1 atm. All simulations were equilibrated for 2 ns, and ran another 10 ns for data analysis.

In the simulations of two -SO$_3$H groups in vapor water, there were 27 H$_2$O (plus 2 H$_3$O$^+$) in a cubic simulation box with edges of

4 × 4 × 4 nm. The -SO$_3$H groups were constrained using the same method described above. $NVT$ simulations were performed. All simulations stared with H$_2$O and H$_3$O$^+$ randomly distributed in the simulation box, and ran for 10 ns to observe the aggregation of water molecules.

In the simulations of water molecules in the iCOF nanosheets. Ten nanosheets stacked in an ABA-fashion, separated by 0.342 nm between layers were placed in the center of a simulation box with edges of 4 × 4.6 × 100 nm. $NVT$ simulations were performed. During all simulations, all atoms of iCOF nanosheets were frozen at their initial positions. All simulations were equilibrated for 2 ns, and ran another 10 ns for data analysis.

### Probability density distribution of hydrated protons
Ab initio molecular dynamics (AIMD) simulations were performed based on density functional theory (DFT) using the Vienna ab initio Simulation Package (VASP)[58] with the projector-augmented wave (PAW)[59] approach applied to describe the interaction between the core and valence electrons. The exchange-correlation interaction was treated within the generalized gradient approximation (GGA) expressed by Perdew-Burke-Ernzerhof (PBE) functional[60]. The Van der Waals interactions were introduced by applying Grimme's correction[61]. Energy cutoff values were set to 400 eV for AIMD simulations and 520 eV for structural optimization. Convergence thresholds for structural optimization were set to $1.0 \times 10^{-5}$ eV per atom in energy and 0.02 eV Å$^{-1}$ in force. Γ point in the Brillouin zone was employed for AIMD simulations in a supercell with a 15 Å vacuum layer. For each sulfonate ligands, seven water molecules were randomly distributed in its vicinity at the initial state. The simulations were taken on the canonical (NVT) ensemble with Nose-Hoover chain thermostat[62], and the time step was set to 1 fs. To prevent flipping of the benzene rings, the z-coordinates of all sulfate atoms were fixed. The following simulation scheme was applied: thermalizing the system to 300 K within 1 ps, equilibrating the structure for 4 ps, and collecting the structural data in a sufficiently long period (40 ps for TpPa-SO$_3$H, 25 ps for TpBd-SO$_3$H). The climbing-image nudged elastic band method (CI-NEB)[63] was employed to calculate the migration barriers for proton hopping between water molecules in different configurations. Excess electrons were compensated with a homogeneous background charge. All the graphs of atomic structures were plotted using VESTA[64], and the probability densities of protons were obtained using Pymatgen[65,66].

## Data availability
All data supporting the findings of this work are available within the article and the Supplementary Information file, or available from the corresponding authors upon request. Source data are provided with this paper.

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

## Acknowledgements

This work was supported by National Natural Science Foundation of China, grant no. 91934302 (Z.J.), 21961142013 (Z.J.), and Program of Introducing Talents of Discipline to Universities, grant no. BP0618007 (Z.J.). The authors greatly acknowledge Prof. Sheng Zhang and Prof. Kaige Zhou from Tianjin University for his valuable suggestions for this work.

## Author contributions

Z.J., H.W., F.P., S.L., and B.S. conceived the idea and designed the research. B.S. and X.P. carried out the experiment. X.Y., X.P., J.S., X.W. C.F., L.C., T.Z., M.Q., Y.K., and Y.L. prepare the COF nanosheets and membranes. S.L., M.Z., and Y.L. carried out density functional theory calculations and classical molecular dynamics simulations. H.W., X.P., F.P., and X.W. provided constructive suggestions for results and discussion. All authors participated in the discussion. B.S., X.P., H.W., S.L., F.P., and Z.J. co-wrote the manuscript.

## Competing interests

The authors declare no competing interests.
