## [Peer Review File · Nature Communications]

REVIEWER COMMENTS

Reviewer #1 (Remarks to the Author):

This paper is meaningful in that it revealed the relationship between the proton conductivity and the H-bond distance, and reported the material with the highest proton conductivity value. I think it is acceptable for publishing in nature communications after the following concerns are fully addressed.

1. There are no Nyquist plots in both main texts and supplementary materials, and only the value of proton conductivity is expressed in dots. For the reliability of the data, Nyquist plots of 6 COFs corresponding to Fig.3 (a) should be included as supplementary materials.
2. In fig.3 (f), it is mentioned that through-plane proton conductivities were measured. However, there is the insufficient description of the method for measuring through-plane conductivity or the sample preparation method. Therefore, a more detailed explanation should be added.
3. In fig.23 of supplementary materials, the conductivity of the tightly packed membrane and the loosely packed membrane were compared. However, there are no experimental evidences that the membrane was made tightly and loosely. Therefore, the authors should present the experimental data supporting the tight and loose membrane fabrications. In addition, in the same figure, it is said that the membrane is made by adjusting the vacuum pressure with high pressure and low pressure. But it seems that the exact pressure value or method should be added.
4. In fig.21 in supplementary materials, there are both square and circle marks for each IEC value. It needs to be corrected which one represents the IEC value.

Reviewer #2 (Remarks to the Author):

Shi et al. report the confinement of short hydrogen bond network ionic covalent-organic framework (COF) surface to achieve ultrahigh proton conductivity. They systematically studied the effect of sulfonic acid group density and distance on proton conductivity and identified the optimal density and distance to realize the maximum proton conductivity. The authors conducted detailed structural characterization and analysis and comprehensive theoretical modeling and simulations to explain the experimental results and gain an in-depth understanding of the mechanisms for enhancement of proton conductivity. This work is very important and provides an effective way to design advanced proton-conductive membranes for energy applications including fuel cells. The manuscript is well written. Therefore, I recommend that the manuscript be accepted for publication after some minor revision. A few minor comments/questions are as follows.

- 1) What is the pore size of the COF used in this study? Is the pore size affected by tailoring the sulfonic acid group density and distance? How important is the pore size for proton conductivity?
- 2) The authors compared the COF with Nafion membrane in terms of proton conductivity. It would be interesting to compare and discuss their fuel cell performance as well although it is not the main focus of this work.

Response to reviewer's comments

Reviewer #1 (Remarks to the Author):

This paper is meaningful in that it revealed the relationship between the proton conductivity and the H-bond distance, and reported the material with the highest proton conductivity value. I think it is acceptable for publishing in nature communications after the following concerns are fully addressed.

Thanks for the highly positive remarks and valuable guidance on our manuscript.

1. There are no Nyquist plots in both main texts and supplementary materials, and only the value of proton conductivity is expressed in dots. For the reliability of the data, Nyquist plots of 6 COFs corresponding to Fig.3 (a) should be included as supplementary materials.

Reply:

Based on the reviewer's valuable guidance, the Nyquist plots of the COFs corresponding to Fig.3 (a) have been supplemented in the revised supplementary materials as shown below:

The original description "The in-plane proton conductivity of the iCOFMs was measured by two electrode method and the proton transfer pathway was illustrated in Fig. 3a." has been revised to "The in-plane proton conductivity of the iCOFMs was measured by two electrode method and calculated according to the resistance (Supplementary Fig.22), and the proton transfer pathway was illustrated in Fig. 3a."

Fig. 22 Nyquist plots of the COFs at 90 °C under 100% RH.

2. In fig.3 (f), it is mentioned that through-plane proton conductivities were measured. However, there is the insufficient description of the method for measuring through-plane conductivity or the sample preparation method. Therefore, a more detailed explanation should be added.

Reply:

Based on the reviewer’s valuable suggestion, the detailed description of the method for measuring through-plane conductivity of the device was supplemented in the Supplementary Materials.

“Proton conductivity of iCOFMs” section in Supplementary Materials

The description “The through-plane proton conductivities of the iCOFMs were measured by sandwiching the iCOFMs between 1M HCl solutions. The samples were prepared by assembling the iCOFMs onto polyethylene diaphragm (100 μm) with pore size of 0.5 mm × 0.5 mm. The resistance with and without iCOFMs was both measured. The resistances of iCOFMs were obtained by removing the resistance of

HCl solution from the total resistance. The through-plane proton conductivities (σ , mS cm^{-1}) of the iCOFMs were calculated by equation (3).” has been added in Supplementary Materials.

Fig. R1 The device for measuring the through-plane proton conductivities of iCOFM.

3. In fig.23 of supplementary materials, the conductivity of the tightly packed membrane and the loosely packed membrane were compared. However, there are no experimental evidences that the membrane was made tightly and loosely. Therefore, the authors should present the experimental data supporting the tight and loose membrane fabrications. In addition, in the same figure, it is said that the membrane is made by adjusting the vacuum pressure with high pressure and low pressure. But it seems that the exact pressure value or method should be added.

Reply:

Based on the reviewer’s valuable guidance, the detailed description for preparing tightly packed membrane and the loosely packed membrane have been supplemented. SEM was utilized to characterize the packed structure of iCOFM and the relevant results were supplied in the revised Supplementary Materials.

The original description “Note that the proton conductivity significantly decreased as the iCOF nanosheets (d~1.0 nm) are loosely packed (Fig. 1d, Supplementary Materials Fig.23)” was revised to “Note that the proton conductivity significantly decreased as the iCOF nanosheets (d~1.0 nm) are loosely packed as characterized by SEM (Fig. 1d,Supplementary Materials Fig.24, Fig.25).”

“Preparation of ionic covalent organic framework membranes (iCOFMs)”

section

The original description “First, the iCOF nanosheets were assembled on the surface of PAN driven by vacuum.” was revised to “First, the iCOF nanosheets were assembled on the surface of PAN driven by vacuum pressure of -0.025MPa and -0.08 MPa.”

Fig.24 SEM images of the cross section of the TpBd-SO₃H prepared by vacuum

pressure of -0.08 MPa (a) and -0.025 MPa (b), respectively.

4. In fig.21 in supplementary materials, there are both square and circle marks for each *IEC* value. It needs to be corrected which one represents the *IEC* value.

Reply:

Based on the reviewer's suggestion, the Fig. 21 in Supplementary Materials was corrected to distinguish the *IEC* value obtained by theoretical calculation and experiment.

“Supplementary Materials”

The original Fig.21 was revised to “

Fig. 21 The *IEC* values of iCOFMs (the square marks represent the theoretical values and the circle marks represent the experimental values).

Reviewer #2 (Remarks to the Author):

Shi et al. report the confinement of short hydrogen bond network ionic covalent-organic framework (COF) surface to achieve ultrahigh proton conductivity. They systematically studied the effect of sulfonic acid group density and distance on proton conductivity and identified the optimal density and distance to realize the maximum proton conductivity. The authors conducted detailed structural characterization and analysis and comprehensive theoretical modeling and simulations to explain the experimental results and gain an in-depth understanding of the mechanisms for enhancement of proton conductivity. This work is very important and provides an effective way to design advanced proton-conductive membranes for energy applications including fuel cells. The manuscript is well written. Therefore, I recommend that the manuscript be accepted for publication after some minor revision. A few minor comments/questions are as follows.

Thanks for the highly positive remarks and valuable guidance on our manuscript.

1) What is the pore size of the COF used in this study? Is the pore size affected by tailoring the sulfonic acid group density and distance? How important is the pore size for proton conductivity?

Reply:

Thank the reviewer for the valuable guidance. According to the chemical structure, simulation results, and XRD patterns, the pore size and group density of TpPa-SO₃H(2), TpBd-SO₃H, TpPa-SO₃H, NATBPa-SO₃H, TFPBPa-SO₃H, CABPTZPa-SO₃H are 1.60 nm, 1.72 nm, 2.30 nm, 2.76 nm, 3.17 nm, and 4.65 nm,

1.29 nm⁻², 0.74 nm⁻², 0.49 nm⁻², 0.325 nm⁻², 0.24 nm⁻² and 0.124 nm⁻², respectively. In this work, the -SO₃H group distance and density were tailored by selecting the ionic amine monomer and aldehyde monomer with different chemical structure. The results demonstrated that iCOFs with low group density and large group distance usually possess large pore size, and *vice versa*. Hence, the pore size of iCOF is affected by tailoring the -SO₃H group density and distance. In addition, as the experiment results demonstrated that large pore size induces low group density and large group distance, resulting in low proton conductivity. In contrast, smaller pore size induces high group density and low group distance, resulting in high proton conductivity.

2) The authors compared the COF with Nafion membrane in terms of proton conductivity. It would be interesting to compare and discuss their fuel cell performance as well although it is not the main focus of this work.

Reply:

Based on the reviewer's instructive suggestions, the fuel cell performance of Nafion117 membrane was measured by the same operating conditions and the results were supplied in revised Supplementary Materials.

The description "Despite the high gas permeability, the iCOFMs exhibit comparable fuel cell performance than that of Nafion 117 under the same condition." has been added in the revised manuscript.

Fig. 31 Single fuel cell performance of the TpBd-SO₃H, TpPa-SO₃H, and Nafion under the same condition.

REVIEWERS' COMMENTS

Reviewer #1 (Remarks to the Author):

The revised manuscript has been improved much, and I recommend publication of the manuscript.

Reviewer #2 (Remarks to the Author):

The authors have addressed the reviewers' comments well. The revised manuscript can now be accepted for publication.

Response to reviewer's coments

Reviewer #1 (Remarks to the Author):

The revised manuscript has been improved much, and I recommend publication of the manuscript.

Reply: Thanks for the positive remark on the manuscript.

Reviewer #2 (Remarks to the Author):

The authors have addressed the reviewers' comments well. The revised manuscript can now be accepted for publication.

Reply: Thanks for the positive remark on the manuscript.